# Novel Mutations Found in Individuals with Adult-Onset Pompe Disease

**DOI:** 10.3390/genes11020135

**Published:** 2020-01-28

**Authors:** May T. Aung-Htut, Kristin A. Ham, Michel C. Tchan, Sue Fletcher, Steve D. Wilton

**Affiliations:** 1Centre for Molecular Medicine and Innovative Therapeutics, Murdoch University, Perth 6150, Australia; m.aung-htut@murdoch.edu.au (M.T.A.-H.); Kristin.Ham@murdoch.edu.au (K.A.H.); s.fletcher@murdoch.edu.au (S.F.); 2Perron Institute for Neurological and Translational Science and The University of Western Australia, Perth 6009, Australia; 3Genetic Medicine, Westmead Hospital, Sydney 2145, Australia; Michel.Tchan@health.nsw.gov.au; 4Sydney Medical School, The University of Sydney, Sydney 2006, Australia

**Keywords:** acid α-glucosidase, adult-onset Pompe disease, *GAA* mutation

## Abstract

Pompe disease, or glycogen storage disease II is a rare, progressive disease leading to skeletal muscle weakness due to deficiency of the acid α-1,4-glucosidase enzyme (GAA). The severity of disease and observed time of onset is subject to the various combinations of heterozygous *GAA* alleles. Here we have characterized two novel mutations: c.2074C>T and c.1910_1918del, and a previously reported c.1082C>G mutation of uncertain clinical significance. These mutations were found in three unrelated patients with adult-onset Pompe disease carrying the common c.-32-13T>G mutation. The c.2074 C>T nonsense mutation has obvious consequences on *GAA* expression but the c.1910_1918del (deletion of 3 amino acids) and c.1082C>G missense variants are more subtle DNA changes with catastrophic consequences on GAA activity. Molecular and clinical analyses from the three patients corresponded with the anticipated pathogenicity of each mutation.

## 1. Introduction

Pompe disease, also known as glycogen storage disease type II (GSD II) (OMIM #232300) [1,2], is caused by mutations in the *GAA* located on chromosome 17q25.2–q25.3 [3]. GSD II prevalence is estimated to be 1 in 5,000 to 10,000 births, depending on the ethnicity and geographic regions, and is inherited in an autosomal recessive manner [4]. The *GAA* encodes the acid α-1,4-glucosidase enzyme (GAA) (EC 3.2.1.3) that breaks down glycogen within the lysosomes. Therefore, a deficiency of GAA activity will result in the accumulation of glycogen in the cell, particularly affecting cellular functions in cardiac and skeletal muscles [5]. 

Depending upon the levels of residual GAA activity, presentation of Pompe disease may vary from the severe form with infantile-onset to a much slower but still progressive juvenile or adult-onset form [6]. The levels of residual GAA enzyme activity present in Pompe patients appears to be the primary determinant for the age onset, the tissues involved (cardiac or not) and the severity of the disease [7,8,9]. In general, less than 1% of GAA activity is found in Pompe patients with severe infantile-onset, whereas some juvenile and most late-onset patients [6] have enzyme levels varying between 2–40% activity. The variation in the residual GAA activity is generally a consequence of various combinations of heterozygous *GAA* alleles that range from null mutations to those with partial activity [10]. 

More than 500 different mutations have been identified in the *GAA* gene to date, and include missense, nonsense, splicing defects, as well as frame-shifting deletions (1 to ~3000 nucleotides), duplications (1 to 17 nucleotides) and gross chromosomal rearrangements (www.pompecenter.nl). One-third of the variants confirmed to be pathogenic show consistent severe phenotype if found in conjunction with another severe mutation. While many nonsense and frame-shifting indels are consistent with a null allele and loss of a functional transcript, some missense variants/mutations also impair function and may partially compromise enzyme activity.

In this paper, we characterized two novel mutations, c.2074C>T, c.1910_1918del, and a previously reported c.1082C>G variant of uncertain clinical significance found in unrelated late-onset Pompe disease patients, also carrying the common c.-32-13T>G *GAA* variant. Since residual GAA activity in the adult-onset condition is presumed to arise solely from incomplete mis-splicing of exon 2 from the allele carrying the c.-32-13T>G mutation, the three novel mutations are predicted to abolish GAA function and would be associated with the severe phenotype if inherited with another null allele. Analysis of both *GAA* transcript and protein from all three patients correlates with the observed phenotypes. 

## 2. Materials and Methods 

### 2.1. Ethics Approvals 

The use of human cells was approved by Murdoch University Human Research Ethics Committee (approval 2013/156) and the Western Sydney Local Health District (WSLHD) Human Research Ethic Committee, Australia (approval HREC/17/WMEAD/358). Patient biopsies were collected after informed consent at the Westmead Hospital. Samples were prepared and analyzed in accordance with the protocols approved by the ethics committees of Murdoch University and WSLHD.

### 2.2. Cell Culture 

All cell culture reagents were purchased from Thermo Fisher Scientific Australia Pty. Ltd. (Scoresby, Australia) and cultures were maintained at 37 °C under a 5% CO_2_/95% air atmosphere unless otherwise stated. Human dermal fibroblasts were propagated in DMEM supplemented with L-glutamine and 10% foetal bovine serum.

### 2.3. Genomic DNA and RNA Extraction 

Genomic DNA was extracted using PureLink^®^ Genomic DNA mini kit (Thermo Fisher Scientific, Scoresby, Australia) according to the manufacturer’s instructions. Total RNA was extracted using MagMax™ nucleic acid isolation kit (Thermo Fisher Scientific, Scoresby, Australia) according to the manufacturer’s instructions incorporating the DNase step included in the kit. Total RNA was assessed using the Nanodrop (ND-1000, Thermo Fisher Scientific, Scoresby, Australia) for quality and quantity. cDNA was synthesised using 125 ng of total RNA, 200 ng of random hexamers (Thermo Fisher Scientific, Scoresby, Australia) and SuperScript^®^ IV reverse transcriptase (Thermo Fisher Scientific, Scoresby, Australia) in 20 µL reactions.

### 2.4. PCR, RT-PCR and qPCR

DNA (20 ng) or cDNA (0.5 µL) was amplified using *TaKaRa LA Taq* DNA polymerase with GC buffer II (Takara Bio USA, Inc., Clayton, Australia), the primers and conditions listed in Table 1. The PCR products were fractionated on 2% agarose gels in Tris-acetate EDTA buffer and images were captured using a Fusion-FX gel documentation system (Vilber Lourmat, Eberhardzell, France). Sequencing was performed at the Australian Genome Research Facility (Perth, Australia) and compared to the reference *GAA* genomic sequence (Accession: NG_009822.1) using BLAST [11]. The qPCR reactions were prepared using fast SYBR™ Green (Thermo Fisher Scientific, Scoresby, Australia), 100 nM (96.5% primer efficiency) and 500 nM (104.9% primer efficiency) primers for *GAA* and *TBP* transcript, respectively and a CFX384 Touch™ Real-Time PCR detection system (Bio-Rad Laboratories Pty., Ltd., Gladesville, Australia). *GAA* (Accession: NM_000152.4) transcript expression relative to the reference transcript *TBP* (Accession: NM_003194.4) was calculated. The expression of the *GAA* transcript relative to *TBP* mRNA was analyzed using the Bio-Rad CFX Manager™ Software Version 3.1 (Bio-Rad Laboratories Pty., Ltd., Gladesville, Australia) and the 2^-ΔΔCT^ method and presented as a fold change compared to healthy control fibroblasts.

### 2.5. Western Blotting

Western blotting was performed on 10 µg of total protein, as determined by Pierce™ BCA protein assay kit (Thermo Fisher Scientific, Scoresby, Australia), fractionated on NuPAGE Novex 4–12% BIS/Tris gels (Thermo Fisher Scientific, Scoresby, Australia) and transferred to Pall FluoroTrans^®^ membranes (Fisher Biotec, Wembley, Australia). The membrane was probed using the antibodies listed in Table 2 for overnight at 4 °C and detected using Luminata Crescendo Western HRP substrate (Merck Millipore, Bayswater, Australia) and Fusion FX system (Vilber Lourmat, Eberhardzell, France).

### 2.6. GAA Enzyme Activity Assay

GAA enzyme activity of dried blood spots was determined using the substrate 4-methylumbelliferyl-α-D-glucopyranoside at the National Referral Laboratory (Adelaide, Australia). To test the activity in cells derived from patients and healthy volunteers, cell pellets were collected by trypsinisation and centrifugation and stored at −80 °C until the assays were performed. The pellets were subjected to three freeze-thaw cycles before resuspension in 50 µL of lysis buffer [10 mM HEPES, 70 mM sucrose, 220 mM mannitol supplemented with 1 x protease inhibitors] and sonicated 6 times for 1 s. The cell lysate was centrifuged at 12,000 ×g for 10 min at 4 °C, the supernatant was collected, and total protein concentration was measured using Pierce™ BCA protein assay kit (Thermo Fisher Scientific, Scoresby, Australia). Approximately 3–5 µg of the total protein lysate in 10 µL volume was used for enzymatic reactions under two pH conditions, 3.9 (for acid-α-glucosidase activity) and 6.5 (for neutral α-glucosidase activity). GAA enzyme activity was initiated by adding 20 µL of 1.4 mM artificial substrate 4-methylumbelliferyl-β-D-glucopyranoside (4-MUG) prepared in two 0.2 M acetate buffers, pH 3.9 and pH 6.5. The reaction was incubated for 1 h at 37 °C before adding 200 µL of stop buffer (0.5 M sodium carbonate, pH 10.7). The fluorescent signals were measured using FLUOstar Omega (BMG LABTECH, Mornington, Australia) with 355 nm excitation and 460 nm emission filters. The ratio of signals generated at pH 3.9 to those at pH 6.5 was calculated. All assays were performed in triplicates.

### 2.7. Measurement of Urinary Tetrasaccharide

Urinary tetrasaccharide levels, normalised to creatinine, were determined by tandem mass spectrometry at the National Referral Laboratory (Adelaide, Australia).

### 2.8. In Silico Predictions

Pathogenicity of the mutations identified in patients was predicted using Mutation Taster (http://mutationtaster.org) [12]. *GAA* gene was used as an input, and transcript ID ENST00000302262 was chosen for all analysis. The results were also compared to the mutations found for the same amino acid reported by others. 

## 3. Results

### 3.1. Clinical Assessment 

Patient 1, a Caucasian with non-consanguineous parents, has a history of progressive weakness of the limbs for more than 10 years since presentation. Although this patient remains independently ambulant at age 54 years, a muscle biopsy showed vacuoles of glycogen. Respiratory symptoms improved significantly on nocturnal bi-level ventilation. Patient 2 is of Syrian ancestry with non-consanguineous parents and an 8-year history of progressive limb-girdle weakness. A muscle biopsy showed sub-sarcolemmal glycogen vacuoles. Patient 9 is Caucasian with non-consanguineous parents and had been first examined at age 16. Hip girdle strength was subtly decreased. Shown in Table 2 is a summary of patient details and clinical assessment.

### 3.2. Molecular Analysis

We established patient and healthy-derived fibroblast cell strains from skin biopsies, propagated as outlined in the materials and methods section, and performed molecular analyses. Since all three Pompe patients were adult-onset, we suspected that these patients might be carrying the common *GAA* mutation, c.-32-13T>G, found in more than two-thirds of adult-onset cases. Cryptic splicing products are characteristic of the c.-32-13T>G mutation [6,7] and was assessed by RT-PCR amplification across *GAA* exons 1 to 5 (Figure 1A) from total RNA extracted from cultured fibroblasts. Cryptic splicing products were observed in all three patients but not in the two healthy samples. We also amplified the remainder of the *GAA* transcript, exons 6 to 20, from all three patients and compared these to those from healthy fibroblasts (Figure 1A) to assess the impact of the second mutations on the *GAA* transcript processing. The amplicons representing the *GAA* transcripts from patient 1 and 2 were consistently less abundant than those generated from the other patients and healthy controls. Apart from this observation, there were no noticeable changes in the sizes of *GAA* transcript amplicons derived from all patients. 

The sequencing of genomic DNA amplicons confirmed the presence of c.-32-13T>G in all patients (Figure 1B). The c.1910_1918del (p.Leu637_Val639del) and c.1082C>G (p.Pro361Arg) mutations were identified from the RT-PCR amplicons from patients 2 and 9, respectively. However, no second mutation was found for patient 1, despite complete sequencing of the overlapping RT-PCR products spanning the *GAA* transcript. The absence of other abnormal *GAA* transcript amplicons and the consistently low levels of the *GAA* amplicons in patient 1 (Figure 1A), led us to speculate that the second mutation could be inducing robust nonsense-mediated decay (NMD) of that transcript. Amplification and sequencing of individual *GAA* exons and flanking intronic sequences from patient 1 genomic DNA, using the primers shown in Table 1, revealed a c.2074C>T (p.Gln692X) mutation in exon 15. 

In addition to identifying mutations in all three patients, we also estimated the levels of full-length *GAA* transcript containing exon 2 by RT-qPCR analysis in all patients and compared these to that observed in the two healthy controls (Figure 1C). Substantial variation in *GAA* transcript levels was evident between the two healthy controls, with healthy control 2 expressing 30% lower levels than that seen in healthy control 1. Patient 1 expressed the lowest amount of full-length *GAA* transcript among all patients, approximately half of the levels of healthy control 2. Since the second *GAA* allele carrying c.2074C>T (p.Gln692X) is subjected to NMD in patient 1, the full-length product detected for this patient is mainly contributed by the transcripts that escaped cryptic splicing, caused by the c.-32-13T>G mutation. 

The amount of full-length *GAA* transcript expressed in patient 2 is similar to that from healthy control 2, while patient 9 had higher levels than those observed in healthy control 1. The full-length *GAA* transcript products amplified for both patients 2 and 9 are transcribed from both alleles, although the majority are contributed by the allele carrying the c.1910_1918del (p.Leu637_Val639del) mutation in patient 2 and c.1082C>G (p.Pro361Arg) in patient 9. 

Both RT-PCR and RT-qPCR analysis showed that there were substantial variations in the level of *GAA* transcript in all three patients. Therefore, we performed enzyme activity assays using protein extracts derived from the patients to determine the residual activities in these patients and correlated to the *GAA* transcript levels (Figure 1D). Protein extracts from healthy control fibroblasts were also included as controls. The GAA activity differed significantly between the healthy individuals, with healthy control 1 having twice the activity of healthy control 2. As anticipated, protein extracts from the patient cultured cells had less than 30% of the activity of healthy control 2. Interestingly, although all three patients expressed different levels of full-length *GAA* transcript, as detected by RT-qPCR, similar levels of GAA enzyme activity were observed for all patients. This result indicated that the higher levels of full-length *GAA* transcript observed in patients 2 and 9 compared to patient 1, arising from the *GAA* allele with the c.1910_1918del (p.Leu637_Val639del) and c.1082C>G (p.Pro361Arg) mutations respectively, did not produce a considerable amount of functional protein. 

To determine whether the observed low GAA activity, despite high levels of *GAA* transcript, in patient 2 and 9 was due to defective production of GAA protein, we analyzed the levels of GAA protein expressed in all patients and control fibroblasts (Figure 1E). We prefer utilising patient-derived fibroblasts over the conventional method of artificially overexpressing mutated protein, as artificially overexpressing mutant protein can introduce confounding factors. Both healthy controls expressed similar quantities of active (70 and 76 kD) and intermediate (95 kD) GAA proteins, and limited levels of precursor protein (110 kD). A protein band migrating slower than 110 kD was also observed in all samples. All three patients’ fibroblasts expressed approximately 50% of the amount of mature active GAA proteins (70 and 76 kD) detected in healthy control 2. In patients 2 and 9, there were low levels of precursor protein (110 kD), which was absent in patient 1. These results, together with the transcript analysis, indicates that the active GAA proteins (70 and 76 kD) found in patient 2 and 9 were translated from the *GAA* transcript that escaped cryptic splicing from to the c.-32-13T>G mutation allele, and the 110 kD precursor protein was translated from the second allele unable to undergo maturation. 

### 3.3. In Silico Analysis

The mutations identified in this study were subjected to *in silico* analysis using Mutation Taster [12]. The gene and transcript ID, *GAA* and ENST00000302262, respectively and the coding sequence positions were selected for the analysis. The outputs of Mutation Taster include DNA and amino acid changes, report of known variants, possible splice site changes, alignment of amino acids to determine conservation and protein features, such as glycosylation.

Not all nonsense mutations may induce efficient NMD, however we confirmed that this was the case for patient 1 during molecular analysis with RT-PCR and DNA sequencing failing to detect this allele. The c.2074C>T (p.Gln692X) *GAA* variant should be pathogenic as the amino acid Gln692 is replaced by a stop codon and the resulting protein will be missing the C terminal 783 amino acids. One predicted consequence of this mutation was induction of abnormal splicing of exon 15 due to possible activation of a donor splice site, with the score increasing from 0.79 to 0.94 at gDNA position 11692. 

The nine base deletion, c.1910_1918del (p.Leu637_Val639del) found in patient 2 would cause an in-frame loss of three amino acids encoded within exon 14, potentially affecting protein feature or possible gain of an acceptor site. Two other pathogenic mutations have been reported for the amino acid Gly638 (substitutions of Val638 or Trp638). All three amino acids deleted, Leu637, Gly638 and Val639, are conserved in murine and chimpanzee GAA protein and since no abnormal splicing was detected, it would appear that the loss of these 3 amino acids seriously compromises GAA activity. Although the software predicted possible activation of a cryptic acceptor site, we did not detect any abnormal splicing in patient 2 mRNA. It was of interest to note that Mutation Taster predicted this was a polymorphism rather than pathogenic variant.

However, the missense change of c.1082C>G (p.Pro361Arg) observed in patient 9 was analyzed using Mutation Taster and predicted to be a disease-causing mutation. The program indicated possible effects on protein features, and substitutions of Pro361 with Leu361 that resulted in severe juvenile-onset Pompe have been reported. This amino acid is conserved in both mouse and chimpanzee and is part of the glycosyl hydrolases family 31. Abnormal *GAA* pre-mRNA splicing was not predicted since the changes in the splice site score were minimal. 

## 4. Discussion

We report three pathogenic mutations c.2074C>T (p.Gln692X), c.1910_1918del (p.Leu637_Val639del) and c.1082C>G (p.Pro361Arg) found in adult-onset Pompe patients carrying the common *GAA* mutation c.-32-13T>G. Deficiency in GAA activity was confirmed in both dried blood spot analyses and *in vitro* assays using protein extracts from dermal fibroblasts derived from patients and healthy individuals. Surprisingly, GAA enzyme activity was found to vary two-fold in the dermal fibroblasts between two healthy controls, although the lower level observed in healthy control 2 was still several fold higher than that found in the three patient derived dermal fibroblasts. 

It has been reported that some 25% of pathogenic missense and nonsense mutations disrupt normal pre-mRNA splicing [13] and consequently RT-PCR studies on the *GAA* mRNA are necessary to fully characterise these mutations. Apart from the c.-32-13T>G variant promoting *GAA* exon 2 skipping, none of the other mutations reported here generate abnormally processed *GAA* transcripts. The c.2074C>T (p.Gln692X) gene lesion is an obvious pathogenic null allele and was not initially detected during the mRNA screen, as this mutation induced robust NMD of the *GAA* transcript variant. Nonsense-mediated decay inhibitor can stabilise a transcript with a nonsense mutation and increase detection of the nonsense mutation [14]. Consequently, any GAA protein observed in patient 1 must have been translated solely from the normal *GAA* mRNA transcripts that escape the abnormal splicing of exon 2 caused by the c.-32-13T>G mutation. 

Frequently, the effect of mutations on protein production, maturation and activity are studied using an *in vitro* system overexpressing the mutant protein. One significant limitation of this system is that overexpression of defective protein may lead to accumulation of misfolded aggregates in the endoplasmic reticulum. Therefore, we prefer to utilise patient-derived cells to study protein activity and expression. Based on our western blotting and GAA activity observations, the in-frame deletion of the c.1910_1918del (p.Leu637_Val639del) is unlikely to produce any functional GAA protein. Similarly, missense mutations of the highly conserved Gly638 have been reported as pathogenic by others [15,16,17,18], with low levels of GAA protein and less than 2% of normal GAA activity in patients with the infantile phenotype, in combination with a null mutation. 

The mutation carried by patient 9 (c.1082C>G, p.Pro361Arg) had been reported by EGL Genetic Diagnostics (Eurofins Clinical Diagnostics, Tucker, GA, USA) (rs755253527), although clinical significance was not provided. A different missense change involving the same amino acid (c.1082C>T, p.Pro361Leu) [19] was reported as pathogenic but postulated to retain some GAA activity. Protein and expression analysis in COS-7 or HEK293T cells, transiently transfected with mutated *GAA* cDNA, showed that the c.1082C>T mutation resulted in lower GAA protein levels [19]. These observations are consistent with the low level of GAA protein observed in cells derived from patient 9.

In summary, we describe three different mutations that severely compromise GAA activity and propose that both c.1910_1918del (p.Leu637_Val639del) and c.1082C>G (p.Pro361Arg) mutations affect both activity and levels of GAA protein, either through inappropriate folding, stability and/or maturation. In addition, we encourage analyses of the consequences of mutation at the molecular level in addition to *in silico* analysis. These analyses can be performed using patient-derived cells such as dermal fibroblasts and myoblasts obtained from skin or muscle biopsies, respectively, or lymphocytes from blood samples. 

## Figures and Tables

**Figure 1 genes-11-00135-f001:**
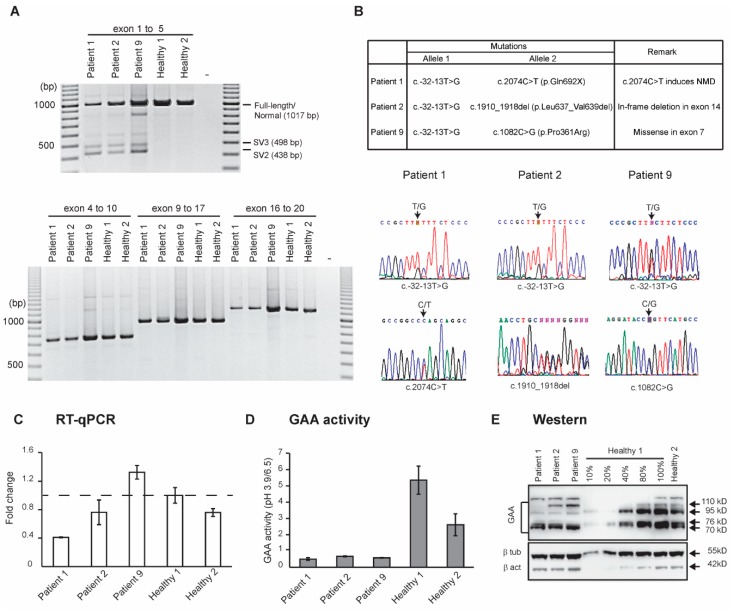
(**A**) RT-PCR analysis of *GAA* transcripts from fibroblasts derived from three Pompe patients and two healthy controls. (**B**) The mutations were identified and confirmed using Sanger sequencing for the three patients. The c.-32-13T>G and c.2074C>T mutations were identified from genomic DNA and the c.1910_1918del and c.1082C>G from RT-PCR products. (**C**) RT-qPCR analysis of the full-length *GAA* transcript containing exon 2. (**D**) GAA activity (N = 3, Error bar = SD) and (**E**) western blotting analysis of GAA (110, 95, 75 and 70 kD) from fibroblasts derived from three patients and two healthy controls.

**Table 1 genes-11-00135-t001:** Primers and antibodies used in this study.

**Primer Names**	**Sequences (5´ to 3´)**	**Purpose**	**Cycling Conditions**
PCR		Genomic DNA amplification data	95 °C for 5 min, 35 cycles of 95 °C 30 s, 60 °C 30 s and 72 °C 1 to 4 min
intron 1F	cagtctagacagcagggcaa		
exon 2R	agtaggatgtgccccaggag		
exon 1F2	cggcctctcagttgggaaa		
exon 2R2	ggttgccaaggacacga		
exon 2F	tgtaggagctgtccaggcc		
exon 5R2	ggcattgctgtttagcag		
intron 4F	gatctcggtcttgaaagc		
exon 10R2	actcagccaccatgtcctcc		
exon 10F	actgccttccccgacttca		
exon 15R	tggaacagtgtgtagaggtg		
exon 15F	cgtacagcttcagcgag		
exon 16R	tgcaggtcgtaccatgtg		
exon 16F	caaggactctagcacctgg		
exon 20R	gaatctcccaagtcctgtgadata		
RT-PCR		*GAA* transcript amplification	95 °C for 5 min 35 cycles of 95 °C 30 s, 60 °C 30 s and 72 °C 1 min
exon 1F	ggaaactgaggcacggagcg		
exon 5R	ggaccacatccatggcattgc		
exon 4F	gtatatcacaggcctcgccg		
exon 10R	ctggtcatggaactcagcca		
exon 9F	gggggttttcatcaccaacga		
exon 17R	ctgccaagggcctctactgg		
exon 16F	caaggactctagcacctgg		
exon 20R	gaatctcccaagtcctgtga		
RT-qPCR		Full-length *GAA* transcript (exon 1-2), amplification	95 °C for 1 min, 40 cycles of 95 °C 3 s, 60 °C 15 s and 72 °C 30 s
exon 1F(q)	tgggaaagctgaggttgtcg		
exon 1-2R(q)	tcctacaggcccgctcc		
		*TBP* transcript amplification	
exon 1-2F(q)	tctttgcagtgacccagcatcac		
exon 2R(q)	cctagagcatctccagcacactct		
**Antibodies**	**Catalogue no.**	**Source**	**Dilution**
rabbit anti-GAA	137068	Abcam, Melbourne, Australia	1:1000
mouse monoclonal anti-β-tubulin	E7	DSHB, Iowa City, Iowa	1: 5000
mouse monoclonal anti-β-actin	A5316	Sigma-Aldrich, Castle Hill NSW	1: 500,000

**Table 2 genes-11-00135-t002:** Clinical assessments of patients.

	Age at Diagnosis	Creatine Kinase (IU/L) (Normal 0–250)	Respiratory Function	Dried Blood Spot GAA Activity (µmol/h/L) (normal 0.3–3.0)	Urine Tetrasaccharides (mmol/mol Creatinine) (Normal <20)
Patient 1	54	320	Type 2 respiratory failure (pCO2 55 mmHg)	<0.1	61
Patient 2	44		Intact	<0.2	150
Patient 9	30	950	Intact	0.8	40

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
