# Peer review of "Novel Mutations Found in Individuals with Adult-Onset Pompe Disease"

_genes, 2020, doi:10.3390/genes11020135_

Round 1
Reviewer 1 Report
The manuscript covers three different variants in the GAA gene that were previously classified as undetermined in addition to the common Caucasian mutation that predicts a late onset variant. The study methodology is solid and very effectively establishes the pathogenicity of these variants. They all produced low GAA enzyme levels and the authors established these were disease causing mutations. I don't have any further edits to this manuscript.
Author Response
Changes in the manuscript
The Materials and Methods and Results sections has been edited as requested by reviewer 2.
Page 2: The instrument used for measuring RNA quantity and quality, and manufacturer are included to clarify in the methods section 2.3 (line 82).
Page 3: The kit used to measure protein concentrations, manufacturer and incubation conditions for antibodies are now included. A new subsection for GAA enzyme activity assays is also added to Materials and Methods section 2.6. The subsection for In silico analysis has been changed to 2.7 and includes additional information.
Page 4: The spelling of Urine tetrasaccharides has been corrected.
Page 5 to 7: We have now expanded the Results section according to the reviewer 2’s recommendations.
Page 9: We have modified the Discussion section.
Reply:
The manuscript covers three different variants in the GAA gene that were previously classified as undetermined in addition to the common Caucasian mutation that predicts a late onset variant. The study methodology is solid and very effectively establishes the pathogenicity of these variants. They all produced low GAA enzyme levels and the authors established these were disease causing mutations. I don't have any further edits to this manuscript.
We thank the reviewer 1 for the time to review the article.
Reviewer 2 Report
Manuscript titled "Novel mutations found in individuals with adult-2
onset Pompe disease' is well written and well thought of experimental case study.
Study reported 3 novel GAA variants found in the background of c.-32-13T>G common LOPD variant. C2074C>T is a straightforward transcript encompassing a null mutation with non-mediated decay of transcript for this allele. c.1910-1918del and c.1082C>G variants have low expression of GAA compared to healthy controls and authors show significantly lower levels of GAA activity. Authors have also used in-silico analysis to predict these variants to be likely pathogenic. However, there are few suggestions to make this manuscript more reader friendly and easy to follow and understand -
Results section is very precise and too short. Characterization of transcripts, expression of all three mutants and their activity are summarized very briefly in a single page, may be a good idea give some more details about their transcript analysis and expression work for readers to understand what and how the experiments were conducted .
Similarly molecular characterization of transcripts and expression of GAA isoforms for each mutant need a bit better description. Authors should not assume that all reader know what they are talking about.
No description of how Alpha glucosidase (GAA) activity was measured, no description of substrate used and no mention of control levels of GAA activity they measured for comparison is described in either methods or result section. In my opinion result section needs to be a little more descriptive and not so abbreviated!
Again, in-silico analysis needs some more elaboration. The variant phenotypes canbe compared with existing pathogenic mutants at genomics and proteomics levels for better clarity and understanding.
Urinine tetrasaccharides – table 2 – needs to be corrected. How were Urine Hex4 measurement done – there is no mention or description of methodology at all.
Author Response
Changes in the manuscript
The Materials and Methods and Results sections has been edited as requested by reviewer 2.
Page 2: The instrument used for measuring RNA quantity and quality, and manufacturer are included to clarify in the methods section 2.3 (line 82).
Page 3: The kit used to measure protein concentrations, manufacturer and incubation conditions for antibodies are now included. A new subsection for GAA enzyme activity assays is also added to Materials and Methods section 2.6. The subsection for In silico analysis has been changed to 2.7 and includes additional information.
Page 4: The spelling of Urine tetrasaccharides has been corrected.
Page 5 to 7: We have now expanded the Results section according to the reviewer 2’s recommendations.
Page 9: We have modified the Discussion section.
Reply:
We thank the reviewer for the constructive feedback and have now made changes according to the recommendations.
Results section is very precise and too short. Characterization of transcripts, expression of all three mutants and their activity are summarized very briefly in a single page, may be a good idea give some more details about their transcript analysis and expression work for readers to understand what and how the experiments were conducted.
Similarly molecular characterization of transcripts and expression of GAA isoforms for each mutant need a bit better description. Authors should not assume that all reader know what they are talking about.
We have modified the Results section with more details on the transcript and protein expression analysis. (Page 5 and 6)
No description of how Alpha glucosidase (GAA) activity was measured, no description of substrate used and no mention of control levels of GAA activity they measured for comparison is described in either methods or result section. In my opinion result section needs to be a little more descriptive and not so abbreviated!
We have now included the description of GAA activity assays performed in the Methods section. (page 3)
Again, in-silico analysis needs some more elaboration. The variant phenotypes can be compared with existing pathogenic mutants at genomics and proteomics levels for better clarity and understanding.
We have now extended the in- silico analysis in result section (page 6 and 7).
Urinine tetrasaccharides – table 2 – needs to be corrected. How were Urine Hex4 measurement done – there is no mention or description of methodology at all.
We have made the correction for Urine tetrasaccharides in table 2 and also described how the measurement were performed.
Reviewer 3 Report
This manuscript describes 3 patients with late onset Pompe disease. All patients have the IVS1 variant on one allele, and another GAA variant on the 2nd allele. The experimental work in this manuscript involves a repetition of standard diagnostic analysis, with the exception of the RT-PCRs, which do not point to aberrant splicing caused by the 2nd variant. In general, it is important to report GAA variants that are associated with Pompe disease and to assess their pathogenic nature. In such reports, it makes sense to collect a decent number of patients and variants in case of standard cases such as reported here before reporting these in the literature in order to prevent a large number of papers reporting only a few variants such as is the case here. To be more specific: the patient and variants reported here in my opinion do not warrant a separate publication. One variant causes a premature stop codon, another causes an in-frame deletion of 3 amino acids, and another variant is a missense variant. There seems nothing special about these variants that warrants their separate publication without any functional analyses. Except for repeating standard diagnostic analysis, the author fail to provide any basic molecular insight in the variants that would justify a separate publication. I recommend submission of these variants to the relevant database for Pompe disease.
Author Response
Changes in the manuscript
The Materials and Methods and Results sections has been edited as requested by reviewer 2.
Page 2: The instrument used for measuring RNA quantity and quality, and manufacturer are included to clarify in the methods section 2.3 (line 82).
Page 3: The kit used to measure protein concentrations, manufacturer and incubation conditions for antibodies are now included. A new subsection for GAA enzyme activity assays is also added to Materials and Methods section 2.6. The subsection for In silico analysis has been changed to 2.7 and includes additional information.
Page 4: The spelling of Urine tetrasaccharides has been corrected.
Page 5 to 7: We have now expanded the Results section according to the reviewer 2’s recommendations.
Page 9: We have modified the Discussion section.
Reply:
This manuscript describes 3 patients with late onset Pompe disease. All patients have the IVS1 variant on one allele, and another GAA variant on the 2nd allele. The experimental work in this manuscript involves a repetition of standard diagnostic analysis, with the exception of the RT-PCRs, which do not point to aberrant splicing caused by the 2nd variant. In general, it is important to report GAA variants that are associated with Pompe disease and to assess their pathogenic nature. In such reports, it makes sense to collect a decent number of patients and variants in case of standard cases such as reported here before reporting these in the literature in order to prevent a large number of papers reporting only a few variants such as is the case here.
We thank the reviewer for the constructive comments. We would also like to draw the reviewer’s attention to the fact that Pompe disease is a rare disease and much less common in Australia with an incidence of one in 145,000 people. There are currently only around 30 people in Australia and apart form the common mutation c.-32-13T>G, we have not identified the same mutation in more than one patient. Even in the Pompe disease GAA variant database, the majority of the mutations are unique with only one patient being reported. Therefore it is not possible to collect a decent number of patients.
To be more specific: the patient and variants reported here in my opinion do not warrant a separate publication. One variant causes a premature stop codon, another causes an in-frame deletion of 3 amino acids, and another variant is a missense variant. There seems nothing special about these variants that warrants their separate publication without any functional analyses. Except for repeating standard diagnostic analysis, the author fail to provide any basic molecular insight in the variants that would justify a separate publication.
The reviewer mentioned that we failed to provide any basic molecular insight of the variants. This study analysed the consequences of each mutation both at the mRNA and protein expression level. Approximately 15% of the GAA mutations account for splicing defects and it is encouraged to confirm if there is any for all mutations. Although we did not study mutated GAA protein using an in vitro system that artificially overexpress GAA protein, patient-derived fibroblasts were utilised to study the consequences of the mutation on GAA protein expression. We prefer to perform phenotypic studies using patient-derived fibroblast so that any influences from an artificial system can be excluded.
Patient 1 (nonsense mutation) has GAA protein translated solely from the normal GAA transcript that escaped the abnormal splicing of exon 2 caused by c.-32-13T>G mutation, and we observed that all three patients express a similar level of GAA protein. Therefore, we concluded that the second mutation found in both patient 2 and 9 affect mutually activity and levels of GAA protein, either through inappropriate folding and/or stability.
I recommend submission of these variants to the relevant database for Pompe disease.
We have already submitted these mutations to ClinVar and will be submitting to Pompe disease database. We believe that providing the consequential evidence for each mutation is crucial and ensuring correct interpretation of the results through peer review is important. Although in silico analysis can be employed to predict the consequences of the mutation, the prediction in fact was incorrect for the mutation found in patient 2. The in-frame deletion was predicted to gain acceptor site/cryptic acceptor sites, however, we did not observe any splicing defect. Therefore we strongly believe that we should publish all novel mutations.
Round 2
Reviewer 2 Report
Please make sure that English language and tenses used are correct. At times it gets hard to follow the thoughts and points presented in the paper.
Author Response
done, see attached file

Reviewer 3 Report
The revised version has not taken away my first response, namely that this work presents standard diagnostics rather than scientific work that merits publication as a work of science. It is important to report small patient numbers of standard diagnostic cases, and for this a database such as the pompe mutation database would be appropriate and sufficient. I don’t see a compelling reason to publish these 3 cases as a separate scientific report as there is nothing exceptional about these cases.
Author Response
done, see attached file
